# Extraction of Microcrystalline Cellulose from Washingtonia Fibre and Its Characterization

**DOI:** 10.3390/polym13183030

**Published:** 2021-09-08

**Authors:** Naved Azum, Mohammad Jawaid, Lau Kia Kian, Anish Khan, Maha Moteb Alotaibi

**Affiliations:** 1Center of Excellence for Advanced Materials Research, King Abdulaziz University, P.O. Box 80203, Jeddah 21589, Saudi Arabia; navedazum@gmail.com (N.A.); anishkhan97@gmail.com (A.K.); 2Chemistry Department, Faculty of Science, King Abdulaziz University, P.O. Box 80203, Jeddah 21589, Saudi Arabia; mmsalotaibi@kau.edu.sa; 3Laboratory of Biocomposite Technology, Institute of Tropical Forestry and Forest Products (INTROP), Universiti Putra Malaysia, Serdang 43400, Malaysia; laukiakian@gmail.com

**Keywords:** microcrystalline cellulose, Washingtonia fibre, morphology, crystallinity, thermal stability

## Abstract

Washingtonia is a desert plant with great sustainability and renewability in nature and is abundantly cultivated across global urban regions. Its fibre biomass comprises cellulose as the major structural part, and this is why it can be potentially utilized as an alternative biomaterial for manufacturing microcrystalline cellulose (MCC) products that can be widely applied in industrial fields. In the present study, NaOH-treated Washingtonia fibre (WAKL), NaClO_2_-treated Washingtonia fibre (WBLH), and Washingtonia microcrystalline cellulose (WMCC) were extracted through combined treatments of alkalization, bleaching, and acidic hydrolysis, respectively. The obtained chemically treated fibre samples were subjected to characterization to investigate their morphology, physico-chemistry, and thermal stability. In a morphological examination, the large bunch WAKL fibre reduced into small size WMCC fibrils, evidencing that the lignin and hemicellulose components were greatly eliminated through chemical dissolution. The elemental composition revealed that almost all impurities of anions and cations had been removed, particularly for the WMCC sample, showing its high purity of cellulose content. Additionally, the WMCC sample could attain at 25% yield, giving it the advantage for feasible economic production. Furthermore, the physicochemical analysis, Fourier Transform Infrared-ray (FTIR), indicated the presence of a crystalline cellulose region within the WMCC structure, which had promoted it with high crystallinity of 72.6% as examined by X-ray diffraction (XRD). As for thermal analysis, WMCC showed greater thermal stability comparing to WAKL and WBLC samples at high temperature. Therefore, Washingtonia fibre can be a reliable biosubstituent to replace other plant material for MCC production in the future.

## 1. Introduction

Exploitation of biomaterials from lignocellulosic biomass is regarded as a green approach for environmental protection. This has received much attention from scientists and researchers to apply it in developing biodegradable composite products. A sustainable environment can be established by utilizing the abundantly available agro-waste originating from plant fibre [1,2,3]. The Washingtonia palm tree is a plant that is commonly cultivated in the hot plain regions across the Middle East, southern United States, southern Europe, and north Africa, which applied for landscaping and ornamental purpose in urban areas [4,5]. However, an excess of agro-waste accumulated from this plant may pose environment issue. The trunk spine part of Washingtonia is normally rich in lignocellulosic contents where the cellulose serves as the main component. This potentially allowed them to act as inexpensive cellulose source for further transformation into smaller sized cellulose products such as microfibrillated cellulose and nanofibrillated cellulose.

Microcrystalline cellulose (MCC) is a product of partial depolymerization of cellulose to level-off the degree of polymerization. Physically, they are generally 50–500 μm in diameter and more than 1 μm in length, and have other appealing properties such as great heat tolerance, biocompatible nature, and high reinforcing capability [6,7]. This has made them widely employed in various industrial fields such as acting as a biofiller for composite reinforcement. Reportedly, numerous studies had been conducted for MCC extraction from various biomass sources by using different chemical methods. Katakojwala and Mohan [1] extracted MCC from sugarcane bagasse via a depolymerization process using both hydrogen peroxide (H_2_O_2_) and sulfuric acid (H_2_SO_4_) after a series of grinding, autoclaving, alkaline (NaOH), and bleaching (NaClO_2_/CH_3_COOH) treatments. Zhao et al. [3] also prepared MCC from tea waste through water boiling, grinding, alkaline, and bleaching (NaClO_2_/HCl/H_2_O_2_) before the hydrolysis process with hydrochloric acid (HCl). Another work by Yavorov et al. [8] studied kinetic hydrolysis by employing dilute acid to isolate MCC from industrial bleached hardwood pulp.

According to the literature, it was reported that Washingtonia fibres could be utilized for biocomposite applications or to derive other cellulosic-based products. Benzannache et al. [9] mechanically improved biocomposite plaster with incorporation of Washingtonian fibres, whilst their performance was examined using a response surface methodology. Another study by Gaagaia et al. [10], who extracted cellulosic fibres from Washingtonia, revealed that the fibres have great thermal stability and tensile strain. However, there is none of work reported on the isolation of MCC from Washingtonia fibre. In previous work, we studied the different biomass parts from Washingtonia plant, including leaf stalk, trunk core, and trunk spine, in relation to their fundamental characteristics and properties [11]. Hence, the novelty of the present study is that is focuses on the extraction of MCC from Washingtonia fibre via combined treatments of alkalization, bleaching, and acidic hydrolysis. The advantage of using the combination of chemical treatments could facilitate the increased yield for MCC product with high purity. Meanwhile, the obtained chemically treated fibres were characterized to study their morphological feature and the sizes of particles. In addition, their chemical compositions, functional groups, cellulose crystalline structures, and thermal behaviours were also examined to comprehensively understand the change in properties.

## 2. Materials and Methods

### 2.1. Materials and Chemicals

Washingtonia trunk spine raw fibre (WTSR) was obtained from Riyadh, Saudi Arabia. The fibre was ground to reduce its sizes to approximately 0.5–1 cm. Before MCC extraction treatment, the fibre was oven-dried at 60 °C for 24 h to further reduce moisture content in reaching between 5 and 7 wt.% using a moisture analyser (Mettler-Toledo International Inc., Columbus, OH, USA). Chemicals of sodium chlorite (NaClO_2_), acetic acid (C_2_H_4_O_2_), hydrochloric acid (HCl), and sodium hydroxide (NaOH) were procured from Evergreen Sdn. Bhd., Malaysia.

### 2.2. MCC Extraction

Initially, about 10 g of grounded fibre was cooked in 600 mL of 2% NaOH at 80 °C for 2 h, with the aim to swell the cellulosic components and remove lignin. The NaOH-treated fibre (WAKL) residue was collected by vacuum filtration through nylon membrane. After that, the fibre further underwent bleaching treatment in 600 mL of 2% NaClO_2_ solution (acidified with 5 mL C_2_H_4_O_2_) at 80 °C for 2 h to further dissolute the lignin and hemicellulose components. The treated solution was then separated through vacuum filtration and the NaClO_2_-treated fibre (WBLH) was collected and washed with distilled water to obtain white colour fibre. Afterwards, acidic hydrolysis was conducted in a 2.5 N HCl solution for 30 min at 80 °C to hydrolyse the amorphous domains of cellulose fibres and isolate the microcrystalline particles (WMCC). About 10 times of cold distilled water was then added to the resulting solution for quenching the acidic reaction. Finally, white pulpy structure of WMCC product was obtained after filtration and drying.

### 2.3. Characterization

#### 2.3.1. Chemical Composition and Yield

The chemical composition of each fibre was analysed to determine their contents of α-cellulose with TAPPI T203cm-99, holocellulose with TAPPI T249-75, and lignin with TAPPI T222 om-88, whilst the hemicellulose was determined by deduction with α-cellulose from holocellulose. In addition, the yield (%) of fibres was also determined with Equation (1) as below:(1)$$Yield\;\left(\%\right)=\frac{M_{2}}{M_{1}}\times 100\%$$
in which *M*_1_: mass of raw fibre; *M*_2_: mass of treated fibre.

#### 2.3.2. Morphological Feature, Particle Size, and Elemental Composition

Morphological features of fibres were investigated using JOEL, JSM-7610F Scanning Electron Microscope (SEM) (JEOL Ltd., Tokyo, Japan), operating at 15 kV voltage. These fibres were coated with a platinum metal layer and then mounted on a carbon-taped stub prior to viewing. Particle size of the fibres was analysed using ImageJ software (National Institutes of Health, Bethesda, MD, USA). Meanwhile, elemental composition of fibres was analysed via the attached Energy Dispersive X-ray (EDX) equipment (JEOL Ltd., Tokyo, Japan) on the microscope.

#### 2.3.3. Fourier Transform Infrared-Ray (FTIR) Spectroscopy

The surface functionality of fibres was examined with Thermo Nicolet Nexus 670 FTIR equipment (Thermo Fisher Scientific Inc., Waltham, MA, USA) in the 4000–600 cm^−1^ wavenumbers range with resolution retained at 4 cm^−1^. The fibres were mixed with potassium bromide (KBr) by grinding and then pelletized prior to analysis.

#### 2.3.4. X-ray Diffraction (XRD)

The crystalline structure of fibres was studied using a PANalytical Empyrean X-ray diffractometer (Malvern Panalytical B.V., Brighton, UK), operating at 45 kV and 40 mA. The fibres were placed on nickel coated steel holder before analysis with of 2°/min scanning rate under Cu Kα radiation. Equation (2) was used to calculate the crystallinity index of fibres following reported work by French and Cintron [12]:(2)$$Crystallinity\;Index,\;CrI\;\left(\%\right)=\frac{\left(I_{200}-I_{am}\right)}{I_{200}}\times 100\%$$
where *I*_200_—the intensity of the peak corresponding to the plane (200) at a Bragg angle of 22.5°; *I_am_*—the intensity of the peak corresponding to the amorphous region, at a Bragg angle of 18.9°.

#### 2.3.5. Thermal Analysis

Heat tolerance behaviour of fibres was evaluated via thermogravimetry analysis (TGA) and its corresponded derivative thermogravimetry (DTG) through a TA-SDT Q600 thermogravimetric equipment (Mettler-Toledo International Inc., Columbus, OH, USA). The analysis was carried out in the 30–900 °C temperature range at 10 °C/min heating rate under nitrogen flowing conditions. Additionally, differential scanning calorimetry (DSC) (Mettler-Toledo International Inc., Columbus, OH, USA) was also conducted in 30–600 °C temperature range with 10 °C/min heating rate to assess the changing thermo-molecular behaviour of fibres.

## 3. Results and Discussion

### 3.1. Morphological, Chemical Composition and Elemental Composition

The SEM micrographs of chemically treated fibres are illustrated in Figure 1. A ruptured feature was observed for the WAKL fibre following the alkali cooking treatment, showcasing the swelling process that happened on the fibre. The WBLH fibre presented a more compactly smooth surface as compared to WAKL. This indicated that residual lignin and hemicellulose were substantially removed at this stage, while the smooth structure of cellulose was still maintained [3,9]. In the later acid hydrolysis treatment, WMCC showed fibrous features since the strong acidic attack of hydronium ions had disintegrated the cellulose structure into individual small size fibrils. According to reported works, the penetration of acidic ions could depolymerize the cellulose by degrading the amorphous regions [8,10]. The diameter and length sizes measured for the WMCC were in the ranges of 60–90 μm and 300–550 μm, respectively. These ranges are smaller than those of the WAKL (200–400 μm and 300–600 μm) and WBLH (300–450 μm and 400–700 μm), proving that the size of fibres was tremendously reduced throughout the chemical processes [1]. Additionally, from Table 1, the yield for WMCC production was able to be maintained at 25.1% after the successive chemical treatments process, along with substantial removal of lignin and hemicellulose, while still preserving a large proportion of α-cellulose content.

In EDX analysis (Figure 2), each fibre showed prominent peaks for carbon and oxygen elements, indicating the classic structure of lignocellulosic biomass. From Table 2, WAKL contains aluminium possibly as a result of the negative charge of hydroxide ions during alkali cooking had accumulated the cation through electrostatic attraction. For WBLH, the presence of bismuth and iodide traces was perhaps due to the collection from soil during harvesting. Meanwhile, the detected platinum element in each fibre sample resulted from the metal coating effect during specimen preparation. However, most of the impurities were eliminated at the end of hydrolysis treatment for the WMCC sample [6]. Hence, this suggested that the combination of alkalization, bleaching, and acidic hydrolysis treatments in this study could produce highly pure MCC with less contaminant residues.

### 3.2. FTIR

Figure 3 presents the FTIR spectra of fibres. All fibre samples showed nearly similar patterns of spectra, implying that the chemical functionality was not significantly influenced by the combined chemical processes. The absorption peaks observed at 3558, 2925, 1639, 1424, 1367, 1093, and 894 cm^−1^ are related to the typical functional groups of cellulosic samples [1,11]. Noticeably, the peak at 3558 cm^−1^ (-OH groups vibration of cellulose) broadened from WAKL to WBLH, and subsequently to WMCC, implying the enhanced cellulose content throughout the treatments [3,12]. Additionally, another peak at 2925 cm^−1^ (C-H stretching of cellulose) became sharper for WMCC. This was due to the increasingly exposed cellulose content within the sample [8,9]. However, there is no significant intensity change at the 1745 cm^−1^ peak (hemicellulose C=O stretching) and 1538 cm^−1^ peak (lignin C=C vibration), indicating that both lignin and hemicellulose compounds were still present in each fibre sample [10,13]. A peak noticed at about 1639 cm^−1^ was correlated to the interaction between cellulose chains and water molecules, and this peak showed reduced sharpness for WMCC. It possibly resulted from the hemicellulose removal, which caused the decrement of peak intensity at 1367 cm^−1^ (CH_2_ bending in cellulose and hemicellulose). Meanwhile, another changing peak intensity was observed at 1424 cm^−1^, which is related to the rearrangement of cellulose segments [2,14]. The peak sharpness at 1093 cm^−1^ (ether groups C–O–C asymmetric vibration) was altered, probably attributed to the cellulose order reorientation. Additionally, the 894 cm^−1^ peak (β-1,4-glycosidic linkages) was noted with increased intensity, which demonstrated the improvement of cellulose contents within samples [6,7].

### 3.3. XRD

Figure 4 presents the XRD patterns of fibre samples. All fibres showed prominent peaks at around 15.3°, 16.8°, 22.5° and 34.7°, reflecting the crystallographic planes of (1–10), (110), (200) and (004) [9,15]. These planes revealed that each fibre had a cellulose I crystalline form with a polymorph of a beta type structure [14,16]. The sharp peak observed at 22.5° was correlated to cellulose crystals domain, and it had the highest intensity for the WMCC product. This signalled WMCC had a stable and strong cellulose structure compared to other fibres [6,13]. Meanwhile, another peak at 15.4° was observed to gradually sharpen for both WBLH and WMCC fibres when compared to WAKL. This was likely due to the bleaching and acid hydrolysis processes that eliminated major residual compounds and ultimately generating high rigid cellulose crystalline structure [2,11]. In addition, the calculated crystallinity for WBLH, 69.4%, was tremendously increased from WAKL, 51.5%. This proved that the bleaching process was highly effective in removing binding components of lignin and hemicellulose after the cellulosic swelling process by alkaline treatment [12,17]. Moreover, the crystallinity only slightly enhanced for WMCC, at 72.6%, probably as a result of the strong acid hydrolysis reaction that had somehow reorganized the crystalline cellulose order in Washingtonia fibre [1,18].

### 3.4. Thermal Analysis

Figure 5 presents the TGA and DTG curves of fibres, while the thermal data are listed in Table 3. All fibre samples showed initial decomposition steps in the temperature range of 70–120 °C. This was promoted by the dehydration process, in which water was removed mostly from the amorphous regions, and while the different mass loss mainly depends on the initial moisture content of the fibres. [6,19]. In the next degradation stage, both WBLH and WMCC samples exhibited greater onset decomposition temperatures (T_OD_) as compared to WAKL. This was possibly promoted by the crystal cellulose structure in bleached and hydrolysed fibres that had largely improved the heat tolerance to higher temperatures [13,16]. Meanwhile, WMCC only revealed a slightly enhanced T_OD_ compared to WBLH, probably due to their similar cellulose crystalline orders [11,20]. In addition, the formation of residual char (W_CR_) was relatively low for both WBLH and WMCC when compared to WAKL. This showcased that the purity of cellulose was improving with the chemical treatment stages [2,12]. With DTG evaluation, the maximal decomposition temperature (T_MD_) increased from WAKL to WMCC samples, evidencing that the thermal stability of fibre samples was enhanced with higher crystallinity degree. The WMCC sample showed the highest T_MD_ at 413.4 °C, and this suggested WMCC is appropriate for use in extreme temperature processing applications [10,21].

From DSC spectra (Figure 6), a wide endotherm extending from 30 to 130 °C was noticeable for all fibres, implying that heat absorption occurred for water evaporation process [7]. At 150–200 °C, a small endothermic band was noted for both WBLH and WMCC at 167.2 and 162.3 °C, respectively. However, this band lacked the significance of that displayed by WAKL at 161.1 °C, while a nearby larger peak occurred at 181.2 °C. This was possibly due to the existence of amorphous components within WAKL that affected their heat flow behaviour [19,21]. Beyond 250 °C, both WBLH and WMCC prominently exhibited the exothermic peak closed to 278.4 °C. However, WAKL sample revealed a shoulder band at 283.6 °C instead of a sharp peak. This indicated that the cellulose decomposition process is much consistent in WBLH and WMCC compared to WAKL fibre. In addition to this, another exotherm at around 345.5 °C was observed for each fibre, in relation to the depolymerization and decarboxylation of cellulose [11]. Near the end, the insignificant endotherms and exotherms appeared after 450 °C, corresponding to the liquefaction and gasification of the cellulose component [9,22].

## 4. Conclusions

The present work revealed findings related to the integrated chemical processes of alkalization, bleaching and acidic hydrolysis with regard to extracting MCC with good properties from Washingtonia fibre. Upon morphological examination, the MCC showed micro-sized fibril structure, which might endow it with great surface reactivity for efficient polymer matrix reinforcement. Furthermore, the surface functionality of all fibres remained unchanged throughout the whole chemical process, whilst most hemicellulose and lignin substances were substantially reduced. Additionally, the crystallinity index of MCC was highly enhanced, suggesting that its strong cellulose crystalline structure is suitable for use in reinforcing applications. In terms of thermal stability, the MCC is capable of withstanding high temperatures due to the highly crystalline structure of cellulose. Therefore, the extracted MCC from Washingtonia trunk spine fibre could be a potential biofiller for developing green composites for various applications in the future.

## Figures and Tables

**Figure 1 polymers-13-03030-f001:**
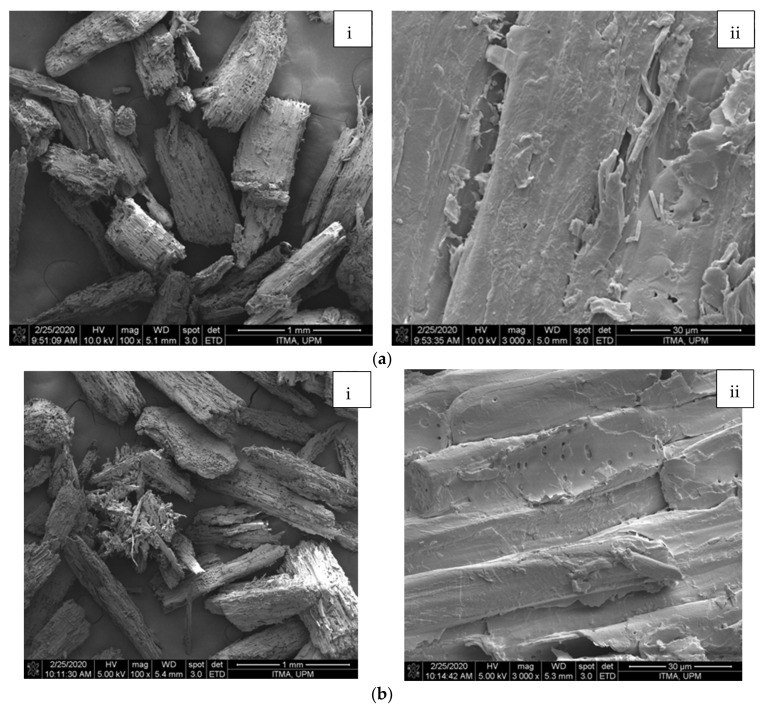
Micrographs of (**a**) WAKL, (**b**) WBLH, and (**c**) WMCC with magnifications of ×100 (i) and ×3000 (ii) under SEM viewing.

**Figure 2 polymers-13-03030-f002:**
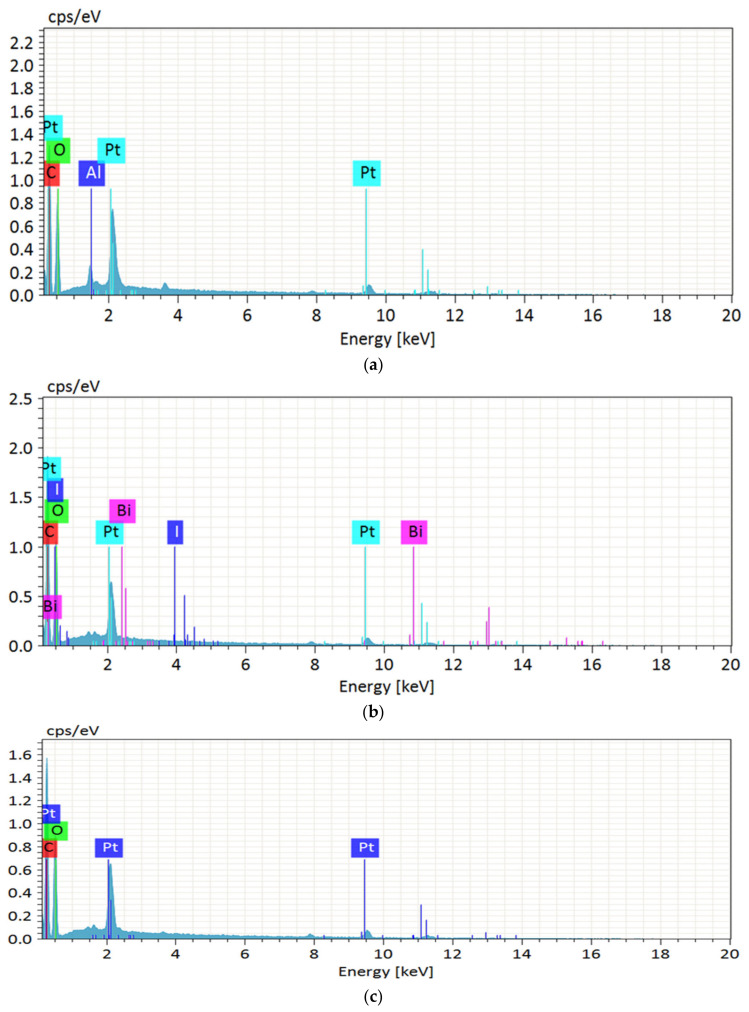
EDX spectra of (**a**) WAKL, (**b**) WBLH, and (**c**) WMCC samples.

**Figure 3 polymers-13-03030-f003:**
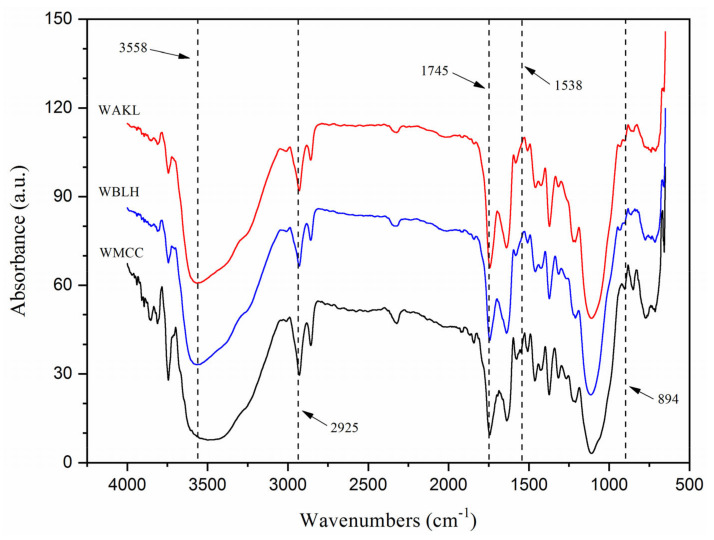
FTIR spectra of WAKL, WBLH and WMCC samples.

**Figure 4 polymers-13-03030-f004:**
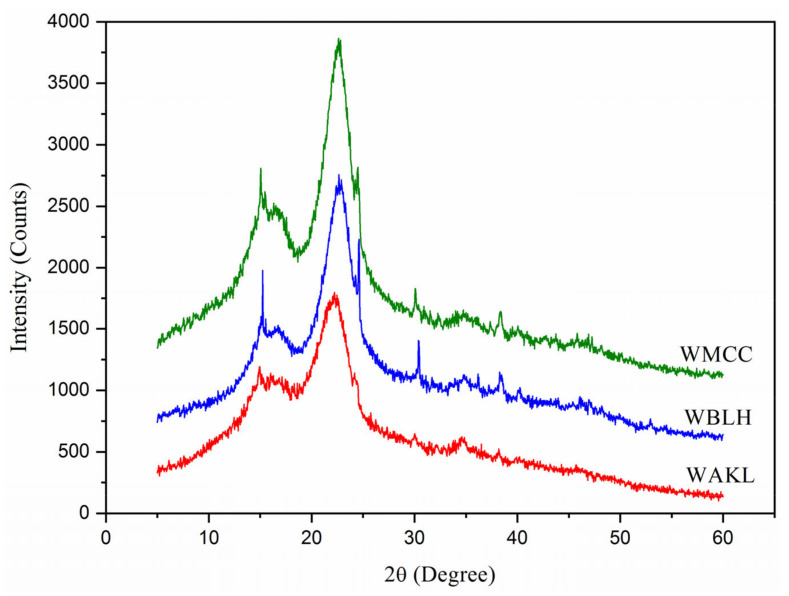
XRD patterns of WAKL, WBLH, and WMCC samples.

**Figure 5 polymers-13-03030-f005:**
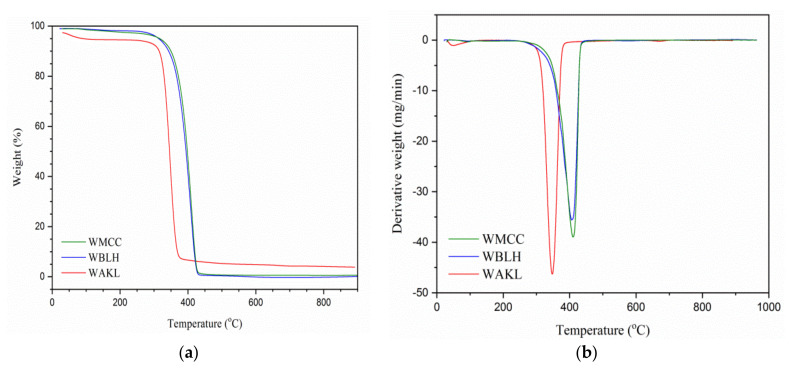
(**a**) TGA and (**b**) DTG graphs of WAKL, WBLH, and WMCC fibres.

**Figure 6 polymers-13-03030-f006:**
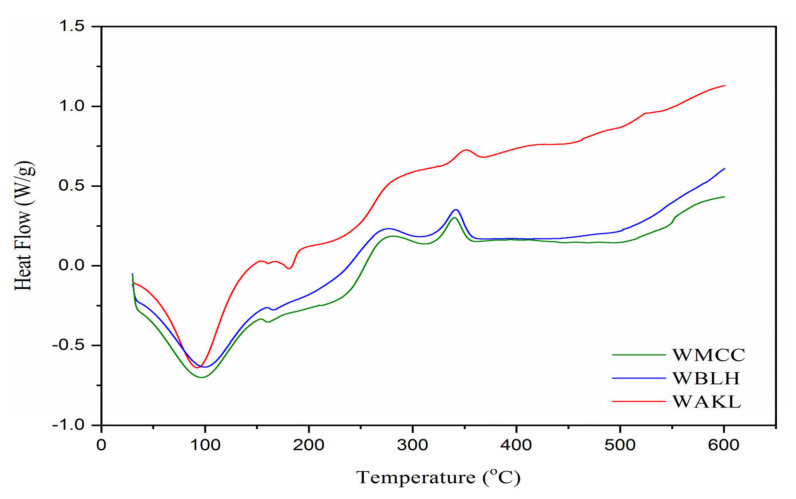
DSC spectra of WAKL, WBLH, and WMCC samples.

**Table 1 polymers-13-03030-t001:** Chemical composition and production yield of fibre samples.

Samples	α-Cellulose (%)	Hemicellulose (%)	Lignin (%)	Yield (%)
WTSR	41.6 ± 0.83	20.9 ± 0.16	21.7 ± 0.17	-
WAKL	72.8 ± 0.53	12.7 ± 0.15	7.4 ± 0.06	46.2 ± 0.35
WBLH	75.1 ± 0.57	9.1 ± 0.17	5.6 ± 0.08	39.4 ± 0.38
WMCC	82.9 ± 0.65	5.2 ± 0.21	2.1 ± 0.13	25.1 ± 0.32

**Table 2 polymers-13-03030-t002:** Elemental composition of fibres.

Samples	C (%) ^a^	O (%) ^b^	Al (%) ^c^	Bi (%) ^d^	I (%) ^e^
WAKL	61.03	37.02	0.89	0.04	0.03
WBLH	67.05	32.11	0.00	0.15	0.22
WMCC	61.55	37.63	0.00	0.00	0.00

Note: ^a^ Carbon; ^b^ oxygen; ^c^ aluminium; ^d^ bismuth; ^e^ iodide.

**Table 3 polymers-13-03030-t003:** Thermal analysis data of WAKL, WBLH and WMCC.

Samples	T_OD_ (°C) ^a^	T_MD_ (°C) ^b^	W_TL_ (%) ^c^	W_CR_ (%) ^d^
WAKL	320.8	346.1	92.5	3.9
WBLH	354.7	401.2	94.7	1.5
WMCC	355.1	413.4	94.3	1.7

Note: ^a^ onset decomposition temperature; ^b^ maximal decomposition temperature; ^c^ total weight loss; ^d^ weight of residual char formation.

## Data Availability

Not applicable.

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
