# Peer review of "Extraction of Microcrystalline Cellulose from Washingtonia Fibre and Its Characterization"

_polymers, 2021, doi:10.3390/polym13183030_

Round 1

Reviewer 1 Report

In this paper, NaOH-treated Washingtonia fibre (WAKL), NaClO2-treated Washingtonia fibre (WBLH), and Washingtonia microcrystalline cellulose (WMCC) were extracted through combined treatments of alkalization, bleaching, and acidic hydrolysis, respectively. Overall, the data of this paper is relatively substantial and the analysis is proper. However, there are many articles that be published in the general area of Cellulose-Fibre Extraction. the paper is not innovative enough. Moreover, the analysis of the reported results is far too descriptive. And it does not provide new insight. So, the paper has some important defects, and does not meet the journal's desired standard. In its present form, I do not recommend this manuscript for publication in Polymers.

Author Response

We revised manuscript as per your Comments

Reviewer 2 Report

Reference paper: Polymers-1310290

I have read the article entitled “Extraction of Microcrystalline Cellulose extracted from Washingtonia Fibre and its characterization” and my comments are summarized below.

Major comment: The manuscript presents the extraction of cellulose and production of microcrystalline cellulose from Washingtonia Fibre. Different characterizations have been performed to determine the physicochemical properties and to confirm the quality of MCC. The paper is of interest to many researchers. The work has apparently been well executed. However, additional information and modifications are necessary to improve the scientific content of article. The paper could be published with major revision.

1– Language should be thoroughly revised as some of the sentences are confusing and some errors can be found.

2–The originality of the paper needs to be further clarified in the introduction part. Only the change of MCC source is not sufficient, since MCC properties have already been investigated, especially in recent years. Highlight the differences between the present and the previous works. For this reason, take into account the following comments.

3–I miss some important references dealing with MCC. Add some recent references to the introduction part to improve the content.

4– How do the authors measure the size of the different fibers obtained by SEM? Were they based on SEM analyses and using a software like ImageJ to measure the size distribution or you have employed dynamic light scattering method. Provide more details about that.

5–How about the presence of platinum in the different samples as given in EDX spectra.

6–The discussion of the FTIR peaks can be improved. For instance the band at 3350-3250 cm–1 show the adsorbed water and indicate the OH stretching vibration of cellulose. Use and cite some recent references to improve the discussion.

7–The XRD and the thermal analysis discussions can be improved to strengthen the paper.

8–Equation 2 is the Segal Method to determine the crystallinity and not that of French et al.

9–The DSC results needs more convincing discussion.

10–Conclusions can improved as well.

Author Response

We revised manuscript as per Comments

Round 2

Reviewer 1 Report

Dear editor: Thank you for inviting me to evaluate the article titled “Extraction of Microcrystalline Cellulose from Washingtonia Fibre and its Characterization”.

In this paper, NaOH-treated Washingtonia fibre (WAKL), NaClO2-treated Washingtonia fibre (WBLH), and Washingtonia microcrystalline cellulose (WMCC) were extracted through combined treatments of alkalization, bleaching, and acidic hydrolysis, respectively. Overall, the data of this paper is relatively substantial and the analysis is proper. The authors have made sufficient modifications according to the modification comments. The manuscript in its present version is apposite for publication in Polymers, and I suggest that this paper be accepted without further modification.

Reviewer 2 Report

The authors have revised their manuscript closely and respond all problem in detail, so i suggest accepting it for publishing.